# Effect of Androgens on Human Fascia

**DOI:** 10.3390/biology14070746

**Published:** 2025-06-23

**Authors:** Caterina Fede, Yunfeng Sun, Xiaoxiao Zhao, Andrea Angelini, Pietro Ruggieri, Carla Stecco

**Affiliations:** 1Institute of Human Anatomy, Department of Neurosciences, University of Padova, via Gabelli 65, 35121 Padova, Italy; yunfeng.sun@studenti.unipd.it (Y.S.); xiaoxiao.zhao@studenti.unipd.it (X.Z.); carla.stecco@unipd.it (C.S.); 2Department of Orthopedics and Orthopedic Oncology, University of Padova, via Giustiniani 2, 35128 Padova, Italy; andrea.angelini@unipd.it (A.A.); pietro.ruggieri@unipd.it (P.R.)

**Keywords:** androgen receptor, testosterone, dihydrotestosterone, sex hormones, fascia, extracellular matrix, collagen

## Abstract

Sex hormones are known to influence connective tissues, but their effects on fasciae remain poorly understood. Particularly, the influence of male hormones on fasciae has remained poorly understood. This study investigated the presence of androgen receptors in human fascia and the impact of dihydrotestosterone (DHT), the active form of testosterone, on collagen production by fascial fibroblasts. Tissue samples from the thoracolumbar fascia and fascia lata of male and female donors revealed androgen receptor expression in both sexes. When treated with different concentrations of DHT corresponding to female and male physiological levels, fibroblasts responded in a dose-dependent manner. At the lower, female-level concentration (0.4 ng/mL), collagen type I significantly increased (from ~2% to 4.8% of the cell area), whereas collagen type III decreased markedly (from ~10.4% to 3.3%), suggesting a shift toward a more fibrotic extracellular matrix. In contrast, higher, male-level concentrations (4–10 ng/mL) induced minimal or no significant changes. These findings indicate that androgens can modulate fascia structure and may help explain greater hormone sensitivity observed in females, whose collagen balance is more responsive to hormonal fluctuations. This has relevant implications for understanding sex differences in tissue mechanics, injury risk, and recovery, and may inform personalized approaches in rehabilitation, sports medicine, and connective tissue disorder management.

## 1. Introduction

Sex hormones—primarily estrogen, progesterone, and androgens—are integral to the regulation of various physiological processes in the human body [1]. Beyond their well-documented roles in reproductive and metabolic functions, emerging evidence underscores their significant influence on connective tissues [2].

Recent studies have also demonstrated that sex hormones directly influence the composition and mechanical properties of fasciae [3]. Research indicates that human fascia expresses estrogen receptor-alpha (ERα) and relaxin receptor 1 (RXFP1), suggesting a direct hormonal effect on fascial cells [4]. These receptors are predominantly localized in fibroblasts, with reduced expression observed in postmenopausal women compared to their premenopausal counterparts, highlighting a potential link between hormonal fluctuations and fascial tissue characteristics [4]. In vitro studies have further elucidated this relationship. Exposure of fascial fibroblasts to β-estradiol resulted in significant alterations in extracellular matrix (ECM) components. Collagen I expression decreased from 6% in the follicular phase to 1.9% in the periovulatory phase, whereas collagen III and fibrillin levels increased, reflecting a shift towards a more elastic ECM composition during ovulation [3]. These changes are consistent with the physiological requirements during periods of increased joint mobility, such as ovulation and pregnancy. The opposite situation happens during menopause, in which the fascia becomes more stiff and less elastic. Moreover, the addition of relaxin-1 consistently promotes a more antifibrotic ECM profile (1.7% of collagen I) [3].

A study employing shear wave elastography revealed that non-users of hormonal contraceptives exhibited greater stiffness in the thoracolumbar fascia compared to users, suggesting that hormonal contraceptives may influence fascial mechanical properties, including stiffness and elasticity [5]. Additionally, the use of aromatase inhibitors, which lower estrogen levels, has been associated with increased joint pain and musculoskeletal discomfort, further implicating estrogen’s role in maintaining connective tissue integrity [6]. In general, musculoskeletal pain and fibrosis are common adverse effects encountered in patients receiving antihormonal treatments, such as tamoxifen for breast cancer patients [7,8].

Testosterone, the primary male sex hormone, also influences connective tissues. It plays a crucial role in the development and maintenance of male secondary sexual characteristics, as well as muscle mass and bone density [9]. Although the direct effects of testosterone on human fascia have not been studied, its effects on connective tissue remodeling and ECM production are well established. A study demonstrated that testosterone treatment reduced the expression of collagen types I and III in vascular smooth muscle cells through the Gas6/Axl signaling pathway [10]. This pathway leads to decreased collagen hyperexpression associated with aging and decreased matrix metalloproteinase-2 (MMP-2) activity. The inhibition of MMP-2 activity by testosterone contributes to the attenuation of ECM remodeling, which is crucial for maintaining tissue integrity and preventing fibrosis [10]. Conversely, testosterone deficiency, common in aging men, is associated with an increased incidence of musculoskeletal disorders, including sarcopenia and osteoarthritis [11,12].

Physiological levels of testosterone attenuate profibrotic activity in cardiac fibroblasts by enhancing nitric oxide (NO) production [13]. Testosterone also inhibits Akt and Smad2/3 phosphorylation via transforming growth factor-β (TGF-β) and angiotensin II (Ang II) pathways, reducing fibroblast activation, migration, and collagen production, potentially benefiting heart failure [14]. In dermal fibroblasts, androgens suppress collagen deposition and modulate inflammation, impacting ECM composition during healing [15].

Testosterone’s effects are mediated by the androgen receptor (AR), either directly or via its more potent metabolite dihydrotestosterone (DHT), produced by 5α-reductase. DHT, which has two to three times greater affinity for AR and influences ECM remodeling by regulating collagen, proteoglycans, MMPs, and TIMPs, which are crucial for tissue homeostasis and repair [16,17]. However, in the context of pathological conditions, such as prostatic fibrosis and desmoid-type fibromatosis, DHT can promote fibrotic responses through AR-dependent mechanisms [18].

Furthermore, several authors have demonstrated that male and female chondrocytes exhibit differential responsiveness to testosterone. Females show higher aromatase activity, favoring conversion to estradiol, whereas in males, DHT mediates sex-specific membrane effects in growth plate chondrocytes [19,20]. These findings highlight the need to understand sex-based differences in connective tissue homeostasis. Hormonal fluctuations, especially in women, can influence musculoskeletal health. For example, variations in fascial stiffness and collagen composition during the menstrual cycle may influence injury risk and recovery outcomes. Hormonal therapies, including hormone replacement therapy and oral contraceptives, may offer therapeutic avenues to modulate fascial properties and alleviate related symptoms [21].

Men typically exhibit greater fascial thickness and tissue stiffness compared to women, likely due to higher basal testosterone levels and increased androgen receptor activity. This sexual dimorphism in connective tissue properties may have biomechanical implications. For instance, males might possess enhanced force transmission capacity through fascia, but potentially reduced elasticity, which could influence susceptibility to certain musculoskeletal injuries or pain syndromes [17]. Furthermore, age-related androgen decline (i.e., andropause) is associated with an increased risk of muscle wasting and joint degeneration, suggesting a protective, maintenance role for testosterone in connective tissue homeostasis [12]. This is supported by evidence of age-related changes in the extracellular matrix (ECM). Fede et al. (2022) reported increased collagen I and reduced elastic fibers and hyaluronan in aged human quadriceps, contributing to muscle stiffness [22]. Similarly, Pavan et al. (2020) found a doubling of intramuscular ECM area and increased passive tension in older adults [23]. These structural changes may be hormonally driven and align with Smith et al. (2010), who emphasized the ECM’s key role in muscle function and mechanical responsiveness [24]. The androgen decline may worsen ECM remodeling, promoting sarcopenia and joint degeneration. On the other hand, testosterone in women, though present in lower concentrations compared to men, contributes to ECM regulation. For instance, studies have shown that testosterone administration can induce muscle fiber hypertrophy and improve capillarization in young women, particularly affecting type II muscle fibers [25]. Furthermore, testosterone inhibits the production of matrix metalloproteinase-1 (MMP-1) in human endometrial stromal cells, suggesting a role in reducing collagen degradation [26].

However, the effects of testosterone are not uniform across all tissues. In the vascular system, for example, testosterone’s influence on endothelial nitric oxide synthase (eNOS) activation differs between men and women [27]. In males, it promotes eNOS activation and NO release via androgen receptor-mediated mechanisms, contributing to vascular relaxation and homeostasis. In contrast, in females, the response to testosterone is more variable and appears to depend on the local hormonal environment, including estrogen levels, which may modulate androgen receptor sensitivity or downstream signaling. These differences contribute to sex-specific vascular responses and may partially explain the divergent cardiovascular effects observed between men and women in both physiological and pathological contexts [27].

Collectively, these findings underscore the importance of testosterone in men and women’s health, particularly concerning connective tissue integrity and musculoskeletal function. Understanding the nuanced roles of testosterone can inform therapeutic strategies aimed at addressing musculoskeletal disorders and age-related changes in connective tissue.

In this work, we aimed to understand whether the androgen receptor is expressed in human deep fascia and how DHT, a potent steroid agonist for the androgen receptor, can influence ECM production by fascial fibroblasts. In this way, we can understand whether male hormones affect the fascia, with the aim to elucidate better gender differences in musculoskeletal health. Continued research is essential to further elucidate how sex hormones affect fasciae and to translate findings into effective clinical strategies for managing connective tissue-related disorders.

## 2. Materials and Methods

### 2.1. Sample Collection

This study was approved by the Ethics Committee of the Hospital of University of Padova (approval no. 5473/AO/22, approval no. AOP2648), and all ethical regulations regarding research conducted on human tissues were carefully followed. Written informed consent was obtained from all the volunteer donors.

Samples of fascia, approximately 0.5 cm × 0.5 cm, were collected from four volunteers (two males and two females patients), with an average age of 64 (range 50–76 y), who were undergoing elective surgical procedures at the Orthopedic Clinic of Padua University. The samples were collected from two fascia lata of the thigh and two thoracolumbar fascia. These samples were transported to the laboratory in phosphate buffered saline (PBS) within a few hours of their collection and were used fresh for cell isolation or were formalin-fixed for histology.

### 2.2. Cell Isolation and Culture

Fresh samples were washed in PBS containing 1% penicillin and streptomycin within a few hours of collection, and were then cut with a surgical scalpel and transferred to tissue flasks with DMEM 1 g/L glucose, 10% FBS, and 1% penicillin–streptomycin antibiotic. Isolated fibroblasts were characterized using anti-Fibroblast Surface Protein (1B10) antibody, as described in one of our previous works [28]. Cell cultures were maintained at 37 °C, 95% humidity, and 5% CO_2_, and were used from passages two to nine.

### 2.3. Immunohistochemistry and Immunocytochemistry for Androgen Receptor

Human fascia specimens were fixed in 10% formalin solution, dehydrated in graded ethanol, embedded in paraffin, and cut into 5 μm-thick sections. To detect AR, dewaxed sections were treated with 1% H_2_O_2_ in PBS for 10 min at room temperature and then washed in PBS. Samples were pre-incubated with a blocking and permeabilizing buffer (1% bovine serum albumin, BSA, and 0.5% Triton-X 0.5%, in PBS), then incubated overnight at 4 °C with a mouse monoclonal anti-androgen receptor antibody diluted 1:100 in PBS–BSA 0.5% (BioCare Medical, Pacheco, CA, USA). After repeated PBS washings, the samples were incubated with anti-mouse IgG, Biotinylated Antibody, 1:600, for 1 h at room temperature, and then in HRP-conjugated Streptavidin (Jackson ImmunoResearch, Cambridgeshire, UK, dilution 1:250) for 30 min. Then, the samples were washed 3 times in PBS. After washing in PBS, the reaction was developed with 3,3′-diaminobenzidine (Liquid DAB + Substrate Chromogen System kit Dako, Carpinteria, CA, USA), stopped with distilled water, and counterstained with hematoxylin. Negative controls were treated similarly, omitting the primary antibody to confirm the specificity of the immunostaining. Slides were then dehydrated and mounted using Eukitt (Agar Scientific, Rotherham, UK).

Isolated cells from fascia were plated (150 cells/mm^2^ in 24-multiwells containing a glass coverslip) and allowed to attach for 48 h at 37 °C. The cells were then fixed by adding 200 µL of 2% paraformaldehyde in PBS (pH 7.4) to each well without washing away the medium, allowing for gentle fixation. After 10 min at room temperature, the cells underwent a second fixation for 10 min with 2% paraformaldehyde in PBS (pH 7.4), were washed three times in PBS, and eventually stored at 4 °C before the staining protocols were carried out. After treatment with 0.5% H_2_O_2_ in PBS for 10 min at room temperature and repeated washings in PBS, the samples were pre-incubated with a blocking and permeabilizing buffer (1% bovine serum albumin, BSA, and 0.2% Triton-X 0.5% in PBS). Afterwards, the same protocol described above was used.

### 2.4. Preparation of DHT Solution

DHT (4,5α-Dihydrotestosterone or 5α-Androstan-17β-ol-3-one) was purchased as an anabolic steroid and AR agonist, derived from testosterone (by Merck, Darmstadt, Germany). Following the instructions, 50 mg of DHT powder was diluted in 1 mL of ethanol solution to prepare a 50 mg/mL stock solution, which was stored at 4 °C before use.

### 2.5. Cell Treatment with Hormone

Isolated cells from the fascia lata and thoracolumbar fascia (from both males and females) were treated with various physiological levels of DHT, as indicated in Table 1. First of all, the cells were plated (150 cells/mm^2^ in 24-multiwells containing a glass coverslip) and allowed to attach for 48 h at 37 °C. The cells were then treated for 24 h with DHT in DMEM without serum, so as not to interfere with the treatment. Endogenous hormone-binding proteins are present in varying concentrations in all serum and plasma samples and may markedly influence hormone treatments and assay results [29]. Various hormone concentrations were used according to the average hormone levels in the blood of males and females. Table 1 reports the mean blood levels of testosterone and DHT in males and females (adolescent vs. adult, and pre- *vs.* post-menopause, respectively). The mean value for testosterone in males is equal to 4 ng/mL, with the maximum reported value equal to 10 ng/mL, whereas in females the mean level before menopause is 0.4 ng/mL, at least 10 times less with respect to males [30,31,32]. DHT values are usually lower than testosterone, with different values in the peripheral tissues, but the plasma levels of the two hormones are highly correlated [33]. Since DHT is a more potent agonist of the AR receptor and is more commonly used in bioassays [34], in this work, it was tested on fascial fibroblasts at concentrations reflecting testosterone levels: 0.4 ng/mL (mean female levels), 4 ng/mL (mean male levels), and 10 ng/mL (high male levels).

In each experiment, one control sample underwent the same processing steps, but did not receive hormones. After treatment, the cells were washed with PBS, gently fixed by adding 200 µL of 2% paraformaldehyde in PBS (pH 7.4) directly to the medium, and then fixed for 10 min with 2% paraformaldehyde in PBS. After repeated washings in PBS, the samples were stored at 4 °C before performing the staining protocols described below.

### 2.6. Collagen Staining

Picrosirius Red staining was first applied to visualize the collagen. Picrosirius Red solution (0.1 g of Sirius Red Sigma Aldrich (St. Louis, MO, USA) (per 100 mL of saturated aqueous solution of picric acid) was applied to the fixed cells for 20 min, and was then washed out with acidified water (5 mL acetic acid, glacial, to 1 L of water) [3].

The immunocytochemistry procedure included the blocking of endogen peroxidase by 0.5% H_2_O_2_ in PBS for 10 min at room temperature, pre-incubation with a blocking buffer (0.1% bovine serum albumin, BSA, in PBS) for 60 min at room temperature, and incubation with the primary antibodies in the same pre-incubation buffer. The antibodies used included goat anti-collagen I (Southern-Biotech (Birmingham, AL, USA), 1:300) and mouse anti-collagen III (Abcam (Cambridge, UK), 1:100). After incubation overnight at 4 °C and repeated PBS washings, the samples were incubated for 1 h with rabbit anti-goat (Jackson ImmunoResearch, 1:300, 1 h) for the detection of collagen I, or with anti-mouse IgG, Biotinylated Antibody, 1:600, for 1 h, followed by a 30-min incubation with HRP-conjugated Streptavidin (Jackson ImmunoResearch, Cambridgeshire, UK, 1:250) for collagen III detection. After 3 washings in PBS, the reaction was developed with 3,3′-diaminobenzidine (Liquid DAB + substrate Chromogen System kit Dako Corp.) and stopped with distilled water, before counterstaining the nuclei with hematoxylin.

### 2.7. Image Analysis

In all samples, nuclei were counterstained with ready-to-use hematoxylin (Dako Corp.). Images were acquired using a Leica DMR microscope (Leica Microsystems, Wetzlar, Germany). Computerized image analysis was performed with ImageJ software 1.54p (freely available at http://rsb.info.nih.gov/ij/, accessed on 17 March 2025) to quantify anti-androgen receptor antibody positivity in the tissue samples, and Picrosirius Red, anti-collagen I, and anti-collagen III antibody positivity in cells (staining was repeated at least 2 times, and for each sample, at least 20 images were analyzed, magnification 20X). The analysis was performed in cells isolated from both the fascia lata and thoracolumbar fascia and is expressed as mean ± standard deviation.

### 2.8. Statistical Analysis

Statistical analyses were performed using IBM SPSS statistical software (version 25, SPSS, Chicago, IL, USA). All data were reported as mean ± standard deviation, and each analysis was replicated at least twice. Data on the percentage of positive area following different doses of DHT were analyzed by One-Way Analysis of Variance (ANOVA), followed by Dunnett’s test for multiple comparisons against the control (untreated) condition. A *p*-value < 0.05 was considered as statistically significant.

## 3. Results

The typical fascial organization of longitudinally oriented collagen fibers and elongated fibroblasts was evident in all the samples (Figure 1 and Figure 2). AR was expressed in the fascial fibroblasts of both the districts analyzed (thoracolumbar fascia and fascia lata) and in all the samples (both males and females) (Figure 1 and Figure 2). Positivity was observed in the fascial fibroblasts (Figure 1 and Figure 2), with no statistically significant difference in reactivity between the two tissues (*p*-value = 0.71): 4.8 ± 2.6% in the fascia lata and 5.2 ± 2.2% in the thoracolumbar fascia. Not all fibroblasts were positive for the AR receptor; some cells remained unstained, which was especially evident at higher magnification (Figure 1D and Figure 2D). No significant gender-related differences were observed using immunohistochemistry in either the fascia lata (Figure 1B,C) or thoracolumbar fascia (Figure 2A,B) (*p*-value = 0.7).

The specificity of the staining was confirmed by the negative control, with the omission of the primary antibody (Figure 1A).

These results were also confirmed in cells isolated from the same fascial tissues (Figure 3), which we characterized as fibroblasts [28], all of which were positive for the hormone receptor (Figure 3C,F). The cells showed a slightly different morphology depending on the fascial district of origin. The cells isolated from fascia lata (Figure 3A–C) were less elongated and had less cytoplasmic extensions with respect to the fibroblasts of the thoracolumbar fascia (Figure 3D–F). The reaction to the androgen receptor antibody showed no statistically significant differences. The positive area was 3.3 ± 1.1% and 3.6 ± 2.1% in the cells of the fascia lata and thoracolumbar fascia, respectively (*p*-value = 0.78). Positivity was nevertheless localized in the cytoplasm of all the cells and also in several nuclei (Figure 3F).

Staining with Picrosirius Red (Figure 4) was used for histological visualization of the collagen fibers synthesized by the cells after exposure to various levels of DHT hormone (Figure 4B–D). The quantification of collagen production, measured as the percentage of positive area, revealed statistically significant modifications compared to control cells (Table 2). After treatment with 4 ng/mL (mean male levels, Figure 4C) and 10 ng/mL (high male levels, Figure 4D), staining decreased from 14.06 ± 3.58% in controls to 11.02 ± 2.7% (*p*-value = 0.048) and 9.77 ± 2.53% (*p*-value = 0.014), respectively (Table 2). The main difference in collagen production was revealed after stimulation with low concentrations of DHT (0.4 ng/mL, female levels) (Figure 4B). The positive area was equal to 9.43 ± 2.29% (*p*-value = 0.012) (Table 2). Neither the treatment nor the absence of serum affected cell density or morphology in the in vitro culture.

Immunocytochemical analysis was performed to explore and quantify the levels of collagen I and collagen III. All results are shown in Figure 5 and Figure 6. Fascial cells modulate the production of the extracellular matrix in response to hormonal stimulation. When treated with a low female-equivalent level of DHT (~0.4 ng/mL, Figure 6B,F) the amount of collagen I significantly increased to 4.80 ± 1.75%, compared to 2.09 ± 0.91% in control cells (*p*-value = 0.0004) (Figure 5). Conversely, collagen III levels were significantly reduced compared with the control (3.32 ± 0.46% of positivity, with respect to 10.46 ± 0.53% of the control cells, *p*-value = 0.0001, Figure 5). In contrast, treatment with male DHT concentrations (4 and 10 pg/mL, Figure 6C,D,G,H) induced less marked changes in collagen production. Specifically, collagen I levels remained similar to those in the control cells (1.98 ± 1.29% for DHT 4 ng/mL and 2.03 ± 0.81% for DHT 10 ng/mL, *p*-value = 0.8, Figure 5). Collagen III did not undergo significant changes at 4 ng/mL (11.19 ± 1.57%, *p*-value = 0.12) and slightly decreased with the higher dose of 10 ng/mL (8.49 ± 1.85%), compared to control levels (10.46 ± 0.53%, *p*-value = 0.015) (Figure 5). In summary, the levels of collagen, both type I and III, remained comparable to the control following exposure of the cells to male doses of DHT. However, exposure to female-level doses of DHT significantly altered extracellular matrix synthesis by fascial fibroblasts, leading to a decrease in collagen III and an increase in collagen I production.

## 4. Discussion

This study provides novel insights into the role of androgens in human fascia, demonstrating for the first time the expression of the AR in fibroblasts derived from both thoracolumbar fascia and fascia lata. This suggests that fascia is a hormonally responsive tissue, capable of modulating extracellular matrix remodeling in response to androgenic signals, as already documented in our previous works on female hormone effects, particularly estrogen and relaxin [3,4,5]. In our previous studies, fascial fibroblasts exhibited significant changes in collagen composition and fibrillin production depending on the menstrual cycle phase and estrogen levels, highlighting the dynamic nature of fascia in women and its sensitivity to hormonal fluctuations. Specifically, estrogen peaks during ovulation or pregnancy reduce collagen type I while increasing collagen type III and fibrillin, promoting a more elastic matrix, whereas the estrogen levels of menopause promote a more fibrotic fascial tissue [3]. However, until now, the direct role of androgens remained poorly understood.

In this study we demonstrated for the first time that human deep fascia is also sensitive to male hormones. Interestingly, we did not observe differences in androgen receptor expression or in the cellular response to hormone treatment based on either the anatomical origin of the tissue (fascia lata vs. thoracolumbar fascia) or the sex of the donor. All DHT concentrations were tested in both female- and male-derived fibroblasts to analyze the dose-dependent responses across sexes. Both groups, regardless of the fascial district, expressed the receptor and responded to dihydrotestosterone in a comparable, dose-dependent manner. Our in vitro experiments revealed dose-dependent modulation of collagen synthesis following DHT treatment. At low (female-equivalent) DHT concentrations, we observed a shift toward a more fibrotic matrix, characterized by an increase in collagen I (4.80 ± 1.75%) and a decrease in collagen III (3.32 ± 0.46%). At male-equivalent concentrations (4 and 10 ng/mL) an overall reduction in total collagen content was observed, decreasing from 14.06% in the control to 11.02% and 9.77% at 4 and 10 ng/mL, respectively. However, the specific levels of collagen I and collagen III, determined by immunohistichemistry, appeared less sensitive at male-equivalent concentrations, with no significant variations compared to the control, except for collagen III at 10 ng/mL (8.49 ± 1.85 vs. 10.46 ± 0.53% in the control). These findings align with previous reports in vascular and dermal fibroblasts, where testosterone and DHT attenuate fibrotic responses by downregulating profibrotic genes and matrix metalloproteinases (MMPs) via AR-mediated signaling pathways [10,13,15,36]. Cutolo and Straub (2020) demonstrated that androgens and progesterone have predominantly anti-inflammatory effects, whereas androgen-to-estrogen conversion is enhanced in inflamed tissues and in patients with autoimmune rheumatic diseases [36]. Moreover, DHT has been shown to modulate the fibroblast phenotype by suppressing TGF-β-induced Smad2/3 phosphorylation, a driver of fibrosis in multiple tissues [14,37].

The distinct response observed at lower female-equivalent androgen concentrations, with enhanced type I collagen production, may reflect a gender difference. In females, DHT may promote ECM stiffening, potentially contributing to injury susceptibility across the menstrual cycle. In contrast, male-range androgen levels appear to support ECM homeostasis by maintaining a balance between collagen synthesis and degradation, thus preventing excessive stiffening (Figure 7).

Interestingly, testosterone deficiency in aging men is associated with increased incidence of sarcopenia, ligament laxity, and joint instability, supporting the hypothesis of the protective role of androgens in connective tissue maintenance [11,12,38,39]. In postmenopausal women, reduced estrogen and androgen levels are correlated with increased stiffness of fascial tissues, which may contribute to the high prevalence of chronic pain syndromes, such as myofascial pain, in this population [6,7,21].

The identification of androgen responsiveness in the fasciae of both sexes has potential clinical implications, for instance, in the role of androgens as modulators in fascial remodeling, in sports injuries, or in surgical recovery. Recent studies in regenerative medicine suggest that testosterone supplementation may enhance tendon and ligament healing by promoting tenocyte proliferation and ECM organization [40]. Additionally, androgen therapies have shown beneficial effects in increasing muscle morphology in young women, resulting in type II fiber hypertrophy and improved capillarization [25], as well as in hypogonadal men, gender-diverse people, and women undergoing gender-affirming hormone therapy [41].

Furthermore, aromatase inhibitors, which suppress estrogen conversion from androgens, are associated with musculoskeletal side effects in breast cancer patients [6,7,8]. Similarly, anabolic steroid abuse—characterized by supraphysiologic androgen levels—may dysregulate ECM homeostasis and contribute to connective tissue fragility and injury [42].

These findings highlight the complex, dose-dependent role of androgens in regulating fascial tissue properties and may help explain sex-based variations in musculoskeletal health by considering hormonal stability across the lifespan. Unlike women, who experience marked fluctuations in sex hormone levels during the menstrual cycle, pregnancy, and menopause, men typically maintain more stable androgen levels throughout adulthood, with a gradual decline during aging. These continuous hormonal fluctuations in the female population have more clinical relevance, leading to cyclic modulation of fascial tension in response to periodic hormone variations.

Future investigations involving a broader range of donor ages, sexes, and hormonal backgrounds may help to clarify specific variations under different physiological or pathological conditions. The in vitro experiments were conducted on cells fixed after 24 h to limit additional variables, such as time. Although longer treatments might reveal additional changes, such as collagen spreading into the extracellular matrix, extending the duration would introduce additional factors and is recommended for future investigations. Additionally, changes in cell proliferation after DHT treatment were not assessed in this study, representing another variable that could influence ECM remodeling and warrants further investigation. Furthermore, the limited number of samples analyzed cannot capture the full extent of the potential biological variability, which represents a limitation of the current work. Moreover, our in vitro experiments cannot fully replicate the complex hormonal and mechanical environment of fasciae in vivo. Nevertheless, this study underscores the importance of androgens in fascial biology and opens new perspectives for research and clinics in the field of musculoskeletal health, with the goal of finding personalized strategies in sports medicine, rehabilitation, and chronic pain management.

## 5. Conclusions

This study provides the first evidence that human fascial fibroblasts express the androgen receptor and are sensitive to stimulation with dihydrotestosterone. Exposure to dihydrotestosterone induced dose-dependent changes in collagen production, with low (female-equivalent) concentrations inducing a shift toward a fibrotic matrix, whereas higher (male-equivalent) concentrations had a lesser impact on the collagen component.

These findings highlight a novel role for androgens in regulating fascial tissue structure and may help explain sex-related differences in connective tissue properties, injury risk, and musculoskeletal health. Recognizing the hormonal responsiveness of fascia could support in the future development of personalized approaches in rehabilitation, injury prevention, and the treatment of myofascial pain disorders.

## Figures and Tables

**Figure 1 biology-14-00746-f001:**
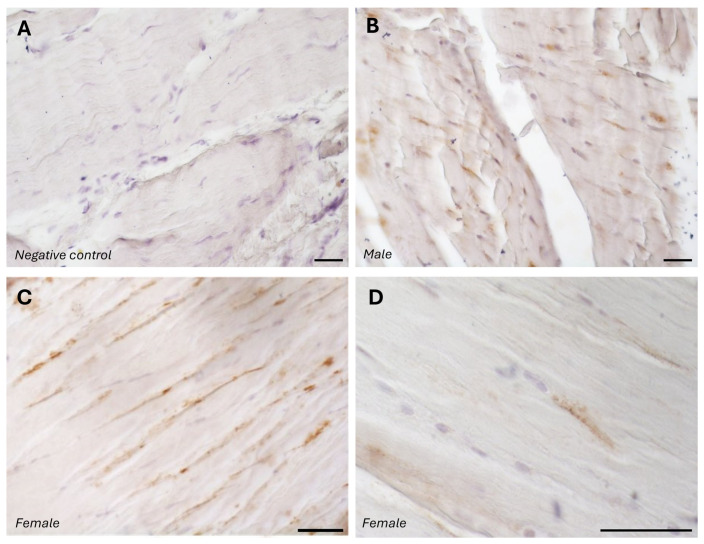
Negative control (**A**) with the omission of the primary antibody, and AR-positive expression in paraffin sections of human fascia lata from a male (**B**) and females (**C**,**D**). Nuclei are counterstained with hematoxylin. Scale bars: 50 µm.

**Figure 2 biology-14-00746-f002:**
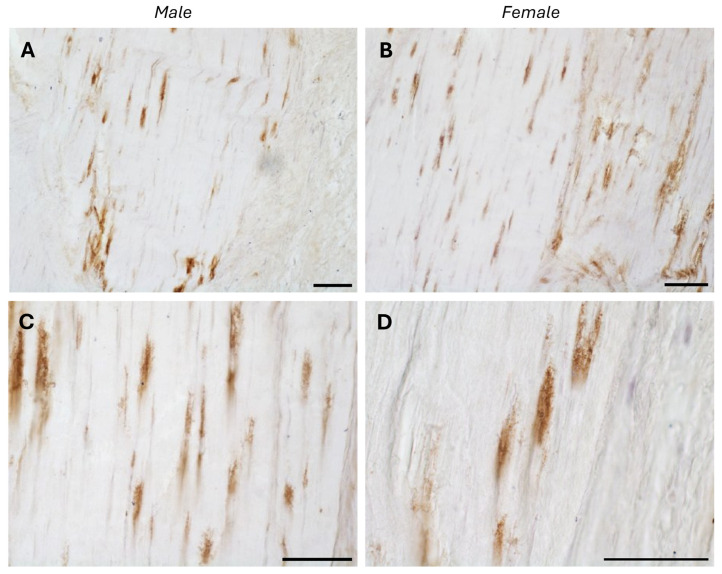
AR-positive expression in paraffin sections of human thoracolumbar fascia in males (**A**,**C**) and females (**B**,**D**). Nuclei are counterstained with hematoxylin. Scale bars: 50 µm.

**Figure 3 biology-14-00746-f003:**
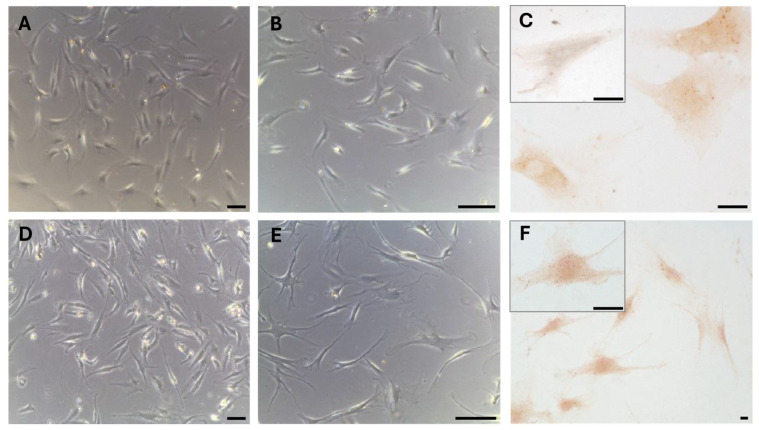
AR expression in fibroblasts isolated from the fascia lata of the thigh (**A**–**C**) and the thoracolumbar fascia (**D**–**F**). Higher magnification images are shown in the insets. Bright-field images are shown in (**A**,**B**,**D**,**E**). Scale bars: 50 μm for A, B, D, and E and 10 μm for C, F and the insets.

**Figure 4 biology-14-00746-f004:**
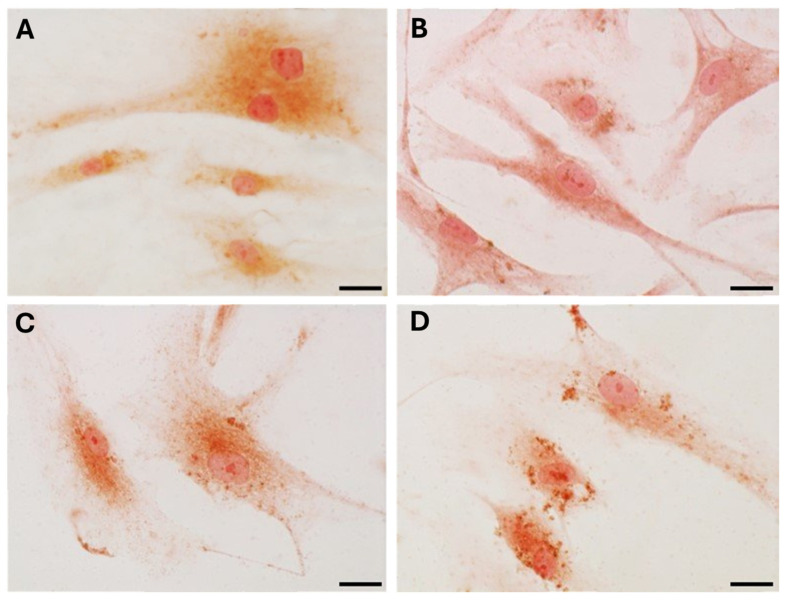
Picrosirius Red staining of fascial fibroblasts after 24 h of treatment with DHT: control cells (**A**), 0.4 ng/mL (**B**), 4 ng/mL (**C**), and 10 ng/mL (**D**). Scale bars: 10 μm.

**Figure 5 biology-14-00746-f005:**
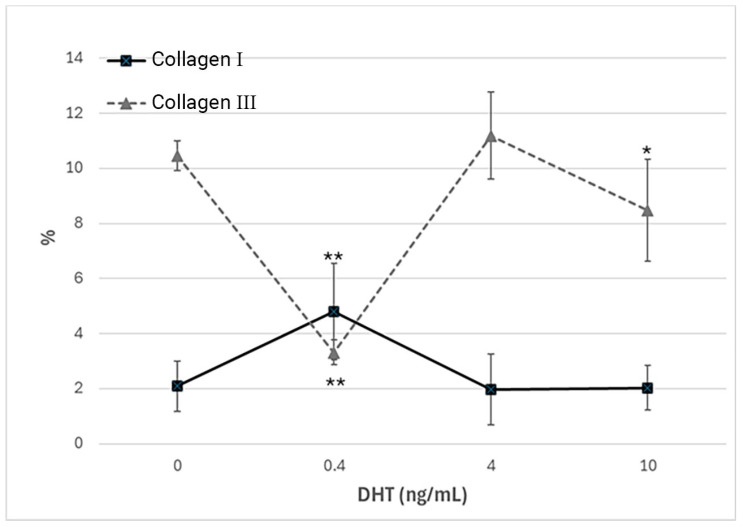
Trend of positivity to anti-collagen I and anti-collagen III, according to the dose of DHT (ng/mL). * *p* < 0.05 (statistically significant) and ** *p* < 0.01 (very statistically significant) with respect to the control, untreated with hormone.

**Figure 6 biology-14-00746-f006:**
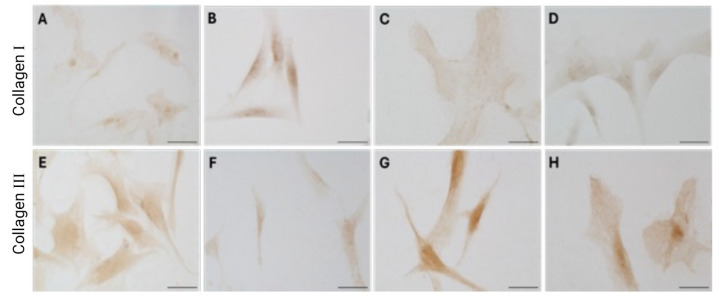
Collagen I (**A**–**D**) and collagen III (**E**–**H**) expression in fascial fibroblasts treated with DHT 0.4 ng/mL (**B**,**F**), 4 ng/mL (**C**,**G**), or 10 ng/mL (**D**,**H**). Control cells (**A**,**E**): cells not incubated with DHT. Scale bars: 50 µm.

**Figure 7 biology-14-00746-f007:**
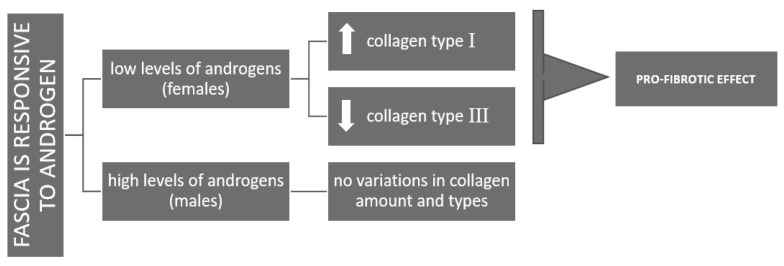
Schemativ representation of the differing responses of fascia to androgens in females (pro-fibrotic effects) and males (homeostasis in ECM production).

**Table 1 biology-14-00746-t001:** Average hormone serum levels in males and females [30,31]. Data are expressed in ng/mL.

	Testosterone	DHT
**Male adolescent**	~3–10 (mean level of 4)	~0.25–1 ^(1)^
**Male adult**	~1–6	~0.25–1 ^(1)^
**Female pre-menopause** **Female post-menopause**	~0.3–0.5 (mean level of 0.4) ~0.25	~0.025–0.4 ~0.01–0.2

^(1)^ In the prostate, the DHT levels are 5–10 times higher [35].

**Table 2 biology-14-00746-t002:** Analysis of collagen content: percentage of positive area after Picrosirius Red staining.

DHT Amount (ng/mL)	%Area (Mean ± SD)
0	14.06 ± 3.58
0.4	9.43 + 2.29 *
4	11.02 + 2.7 *
10	9.77 ± 2.53 *

* *p* < 0.05, statistically significant with respect to the control.

## Data Availability

The original contributions presented in this study are included in the article. Further inquiries can be directed to the corresponding author.

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
