# Peer review of "Effect of Androgens on Human Fascia"

_biology, 2025, doi:10.3390/biology14070746_

Round 1
Reviewer 1 Report
Comments and Suggestions for Authors
This is a novel and interesting study, with high significance to many individuals undergoing androgen enhancing or reducing treatments for various reasons including prostate cancer treatment, and individuals with age-induced reductions in androgens.
This study is well designed and delivered. Reading it was a pleasure.
I have a few small / minor comments for improvement - related to the use of "stiffness" and "muscle wasting", the need for bit more clarification of immunohistochemistry image analysis. And a few small questions.
Are tissues and cells from four donors enough to capture potential biological variablity? Or should this be a limitation of the current study. This question does not mean more work should be added. Just a consideration for the discussion.
More links between muscle wasting/sarcopenia and connective tissue changes within intramuscular connective tissues need to be added. The authors have written about this before, yet readers may need this important link solidified. Cite author's own findings on this topic, as well as Alec S.T. Smith, Rishma Shah, Nigel P. Hunt, Mark P. Lewis,
The Role of Connective Tissue and Extracellular Matrix Signaling in Controlling Muscle Development, Function, and Response to Mechanical Forces (and others as appropriate).
A bit of caution is needed with the term "stiffness". The authors did not test stiffness here directly. Also, stiffness is not just a result of collagen type I levels. Either alter the word to "fibrotic or pro-fibrotic", as done in Figure 7, or add a bit more to the discussion of other potential causes of "stiffness". This include collagen organization and architecture.
Small questions -
1) In tissues, did the % area with collagen take into account the spread of collagen into tissues (since it is a deposited matricellular protein)? Or was the image analysis only within cells in the tissues. While this was easily done for the in vitro cell analysis (as shown clearly in figures 3 and 6), how can one differentiate just cellular collagen in the tissue sections? Perhaps I am just overreading the descriptions in the methods and results.
2) In the cell studies, were proliferative changes also quantified? If not, this might be nice for a future study.
3) In the cell studies, were any studies taken further than 24 hours? While not necessary here - and perhaps not wise since time would be an additional variable - this might be nice for a future study since over time, collagen deposition and picrosirius red staining would expand past the cytoplasm and into the surrousding matrix.
Author Response
This is a novel and interesting study, with high significance to many individuals undergoing androgen enhancing or reducing treatments for various reasons including prostate cancer treatment, and individuals with age-induced reductions in androgens.
This study is well designed and delivered. Reading it was a pleasure.
Thank you for the feedback and the positive comment.
I have a few small / minor comments for improvement - related to the use of "stiffness" and "muscle wasting", the need for bit more clarification of immunohistochemistry image analysis. And a few small questions.
Thank you, we are pleased to improve the manuscript based on your comments.
Are tissues and cells from four donors enough to capture potential biological variablity? Or should this be a limitation of the current study. This question does not mean more work should be added. Just a consideration for the discussion.
This is a thoughtful comment. We believe that the positive expression observed in all four donors—who differ in both sex and anatomical site—provides good evidence that fasciae respond to testosterone. However, we agree that the full extent of potential and complex biological variability cannot be captured with only four samples. Therefore, we have included this point as a limitation of the study (lines 524-539): “Future investigations involving a broader range of donor ages, sexes, and hormonal backgrounds may help to clarify specific variations under different physiological or pathological conditions. The in vitro experiments were conducted on cells fixed after 24 hours to limit additional variables like time. While longer treatments might reveal additional changes, like collagen spreading into the extracellular matrix, extending time points would involve additional factors and is recommended for future investigations. Additionally, changes in cell proliferation after DHT treatment were not assessed in this study, representing another variable that could influence ECM remodeling and need further studies to be investigated. Furthermore, the limited number of samples analysed cannot capture the full extent of the potential biological variability, which represents a limitation of the current work. Moreover, our in vitro experiments cannot fully replicate the complex hormonal and mechanical environment of fasciae in vivo. Nevertheless, this study underscores the importance of androgens in fascial biology and opens new perspectives for research and clinics in the field of musculoskeletal health, with the goal of finding personalized strategies in sports medicine, rehabilitation, and chronic pain management.”
More links between muscle wasting/sarcopenia and connective tissue changes within intramuscular connective tissues need to be added. The authors have written about this before, yet readers may need this important link solidified. Cite author's own findings on this topic, as well as Alec S.T. Smith, Rishma Shah, Nigel P. Hunt, Mark P. Lewis, The Role of Connective Tissue and Extracellular Matrix Signaling in Controlling Muscle Development, Function, and Response to Mechanical Forces (and others as appropriate).
We thank the reviewer for the good comment. We have integrated our text with 3 more references. The revised new text is (lines 196-203): “Furthermore, age-related androgen decline (i.e., andropause) is associated with increased risk of muscle wasting and joint degeneration, suggesting a protective, maintenance role for testosterone in connective tissue homeostasis [12].​ This is supported by evidence of age-related changes in the extracellular matrix (ECM): Fede et al. (2022) reported increased collagen I and reduced elastic fibers and hyaluronan in aged human quadriceps, contributing to muscle stiffness [22]. Pavan et al. (2020) similarly found a doubling of intramuscular ECM area and increased passive tension in older adults [23]. These structural changes may be hormonally driven and align with Smith et al. (2010), who emphasized the ECM’s key role in muscle function and mechanical responsiveness [24]. The androgen decline may worsen ECM remodeling, promoting sarcopenia and joint degeneration.”
A bit of caution is needed with the term "stiffness". The authors did not test stiffness here directly. Also, stiffness is not just a result of collagen type I levels. Either alter the word to "fibrotic or pro-fibrotic", as done in Figure 7, or add a bit more to the discussion of other potential causes of "stiffness". This include collagen organization and architecture.
We totally agree with this observation. We have changed the term “stiffer” in “fibrotic” (both in the summary and in the Discussion, lines 469 and 545).
Small questions -
- In tissues, did the % area with collagen take into account the spread of collagen into tissues (since it is a deposited matricellular protein)? Or was the image analysis only within cells in the tissues. While this was easily done for the in vitro cell analysis (as shown clearly in figures 3 and 6), how can one differentiate just cellular collagen in the tissue sections? Perhaps I am just overreading the descriptions in the methods and results.
Thank you very much for your insightful comment. In the image analysis of tissue sections, the percentage area occupied by collagen was calculated considering the entire area of interest (the brown staining after DAB reaction), thus including both the collagen deposited in the extracellular matrix and any collagen potentially present within the cells.
It is important to highlight that, unlike in vitro analyses where cells are isolated and clearly defined, in tissue sections it is very challenging to distinguish precisely between “intracellular” and “extracellular” collagen, since collagen is primarily a deposited matricellular protein located mainly in the extracellular space.
For this reason, the analysis was performed on the entire observed tissue area, reflecting the overall distribution of collagen within the tissue microenvironment.
We hope this explanation clarifies the point.
- In the cell studies, were proliferative changes also quantified? If not, this might be nice for a future study.
In the current study, we did not quantify proliferative changes in the cell experiments. Moreover, the cells were fixed after 24 hours, a time frame that may be short to fully capture changes in cell viability or proliferation. However, we agree that assessing proliferation would provide valuable additional information, and we consider it an excellent aspect to explore in future research. We appreciate your recommendation. We will take it into account for subsequent studies, and we have added a brief comment regarding this point in the Discussion (Lines 524-539): “Future investigations involving a broader range of donor ages, sexes, and hormonal backgrounds may help to clarify specific variations under different physiological or pathological conditions. The in vitro experiments were conducted on cells fixed after 24 hours to limit additional variables like time. While longer treatments might reveal additional changes, like collagen spreading into the extracellular matrix, extending time points would involve additional factors and is recommended for future investigations. Additionally, changes in cell proliferation after DHT treatment were not assessed in this study, representing another variable that could influence ECM remodeling and need further studies to be investigated. Furthermore, the limited number of samples analysed cannot capture the full extent of the potential biological variability, which represents a limitation of the current work. Moreover, our in vitro experiments cannot fully replicate the complex hormonal and mechanical environment of fasciae in vivo. Nevertheless, this study underscores the importance of androgens in fascial biology and opens new perspectives for research and clinics in the field of musculoskeletal health, with the goal of finding personalized strategies in sports medicine, rehabilitation, and chronic pain management.”
- In the cell studies, were any studies taken further than 24 hours? While not necessary here - and perhaps not wise since time would be an additional variable - this might be nice for a future study since over time, collagen deposition and picrosirius red staining would expand past the cytoplasm and into the surrousding matrix.
As mentioned before, in our current study the cells were fixed after 24 hours—a timeframe chosen deliberately to limit the introduction of additional variables while allowing us to assess collagen production. While extending the duration of the cell experiments beyond 24 hours was not necessary for this study, we agree that longer-term analysis could provide valuable insights. Over extended periods, collagen deposition can evolve, and picrosirius red staining may expand beyond the cytoplasm into the surrounding extracellular matrix, which could be very informative as long-term studies. We will take this into consideration and we have added a brief comment regarding this point in the Discussion (Lines 524-539), as already reported above.
Reviewer 2 Report
Comments and Suggestions for Authors
Dear authors;
Thank you for your valuable manuscript and I enjoyed reading your text. Below I provide my commentaries.
Abstract:
- The simple summary consists of no more than 200 words in one paragraph.
- Revise your first sentence. There are several articles on this issue.
- Write your control group and test duration.
Introduction:
- Line 68, remove the double space.
- Your introduction is too long. Line 102-116 can be removed and summarize line 117-141.
Materials and Methods:
- Line 234, please write your groups first.
- Did you have any references for collagen staining? Please add it.
- Which software did you use for statistical analysis? Write it in the text. Also write about mean ± standard deviation in this section.
- Please specify the number of replicates.
Results:
- Figure 2, the scale bars are not the same. In addition, please add the terms "female" or "male" at the top of the images. You can do it for other figures, too.
- Write the p values in the results.
- Check the scale bars in all figures.
- It would be better to show the changes after treatment using a graph.
- Did you check all concentration in both female and male groups? This can help you reach better conclusions.
Author Response
Thank you for your valuable manuscript and I enjoyed reading your text. Below I provide my commentaries.
Thank you for the positive feedback. We are pleased to improve the manuscript based on your comments.
Abstract:
The simple summary consists of no more than 200 words in one paragraph.
Thank you for the comment: we have revised the summary. The new version is” Sex hormones are known to influence connective tissues, but their effects on fasciae remain poorly understood. Particularly, the influence of male hormones on fasciae has remained poorly understood. This study investigated the presence of androgen receptors in human fascia and the impact of dihydrotestosterone (DHT), the active form of testosterone, on collagen production by fascial fibroblasts. Tissue samples from thoracolumbar fascia and fascia lata of male and female donors revealed androgen receptor expression in both sexes. When treated with different concentrations of DHT corresponding to female and male physiological levels, fibroblasts responded in a dose-dependent manner. At the lower, female-level concentration (0.4 ng/mL), collagen-type I significantly increased (from ~2% to 4.8% of cell area), while collagen-type III decreased markedly (from ~10.4% to 3.3%), suggesting a shift toward a more fibrotic extracellular matrix. In contrast, higher, male-level concentrations (4–10 ng/mL) induced minimal or no significant changes. These findings indicate that androgens can modulate fascia structure and may help explain greater hormone sensitivity in females, whose collagen balance is more responsive to hormonal fluctuations. This has relevant implications for understanding sex differences in tissue mechanics, injury risk, and recovery, and may inform personalized approaches in rehabilitation, sports medicine, and connective tissue disorder management.”
Revise your first sentence. There are several articles on this issue.
We have revised following the right suggestion. The modifed sentence is “Androgens are emerging as important regulators of connective tissue remodeling, but current knowledge about their role in human fascia is still limited”.
Write your control group and test duration.
Thank you for the indications. We have added the requested info. The revised paragraph is: “AR expression was assessed by immunohistochemistry and immunocytochemistry. Fascial fibroblasts were treated in vitro for 24h with DHT at concentrations reflecting physiological levels: 0.4 ng/mL (female), 4 ng/mL (male average), and 10 ng/mL (high male dose). Collagen content was quantified using Picrosirius Red staining, and Collagen-I and III were evaluated by immunocytochemistry and image analysis and compared to the control group not treated.”
Introduction:
Line 68, remove the double space.
Thank you for the indication, but there isn’t a double space.
Your introduction is too long. Line 102-116 can be removed and summarize line 117-141.
We reduced the entire indicated section to less than half of its original length (lines 124-146). Thank you for the suggestion
Materials and Methods:
Line 234, please write your groups first.
Corrected: “Isolated cells from fascia lata and thoracolumbar fascia (from both male and female) were treated with various physiological levels of DHT, as indicated in Table 1. First of all, cells were plated (150 cells/mm2 in 24-multiwells containing a glass coverslip) and allowed to attach for 48 h at 37°C.”
Did you have any references for collagen staining? Please add it.
We have added the reference number 3.
Which software did you use for statistical analysis? Write it in the text. Also write about mean ± standard deviation in this section. Please specify the number of replicates.
The paragraph related to the statistical analysis was entirely revised thanks to the reviever’s suggestion. The new version is: “2.8 Statistical analysis: Statistical analyses were performed by using IBM SPSS statistical software (version 25, SPSS, Chicago, IL, USA). All data were reported as mean ± standard deviation, and each analysis was replicated at least twice. Data of percentage of positive area following different doses of DHT were analyzed by One-Way Analysis of Variance (ANOVA), followed by Dunnett’s test for multiple comparisons to the control (untreated) condition. p < 0.05 was always considered as the limit for statistical significance.“
Results:
Figure 2, the scale bars are not the same.
Yes, the scale bars are not the same because the pictures indicate different enlargement. But, as indicated in the Figure legend, all the scale bars are equal to 50 µm.
In addition, please add the terms "female" or "male" at the top of the images. You can do it for other figures, too.
Thank you for the good suggestion: we have added the terms in both Figure 1 and 2.
Write the p values in the results.
Thank you for the right indication: we now have added all the p-values, at lines 359-364-386-401-404-420-422-425-427 and 429.
Check the scale bars in all figures.
As indicated above. We checked all the scale bars. Thank you for the indication.
It would be better to show the changes after treatment using a graph.
We have already prepared graphs illustrating the changes after treatment for both Collagen I and Collagen III (Figure 5). Regarding total collagen, although testosterone appears to have a relevant effect compared to the control, no clear dose-dependent differences were observed. For this reason, we have only reported the data in the table (Table 2).
Did you check all concentration in both female and male groups? This can help you reach better conclusions.
Yes, we did it. We checked all concentrations in both female and male groups to ensure a comprehensive analysis and to support more robust conclusions. Thank you for the comment, we have added this point in the Materials and Methods (Line 287) and in the Discussion. (Lines 463-467).